# Effects of HfB_2_ Content and Microwave Sintering on the Mechanical Properties of Ti_2_AlC Composites

**DOI:** 10.3390/ma18122693

**Published:** 2025-06-07

**Authors:** Matheus dos Santos Dias Damaceno, Yuri Alexandre Meyer, Eder Lopes Ortiz, Giovana da Silva Padilha, Wislei Riuper Osório, Ausdinir Danilo Bortolozo

**Affiliations:** 1Centro de Pesquisa de Manufatura de Materiais Avançados—CPMMA, Faculdade de Ciências Aplicadas, Universidade Estadual de Campinas (Unicamp), Limeira 13483-350, SP, Brazil; msd.damaceno@gmail.com (M.d.S.D.D.); giovanap@unicamp.br (G.d.S.P.); wislei1@unicamp.br (W.R.O.); 2Faculdade de Tecnologia, Universidade Estadual de Campinas (Unicamp), Limeira 13484-332, SP, Brazil; yameyer@unicamp.br (Y.A.M.); ederlopesortiz@gmail.com (E.L.O.)

**Keywords:** Ti_2_AlC, microwave heating, composite, mechanical properties

## Abstract

This study investigates the influence of the HfB_2_ content and sintering method on the mechanical behavior of Ti_2_AlC-based composites. Compositions containing 0–10 wt.% HfB_2_ are processed via conventional and microwave sintering at 1200 °C for 30 min. X-ray diffraction and scanning electron microscopy analyses have confirmed the formation of the Ti_2_AlC and HfB_2_ phases, whereas the TiC phase is predominantly observed in samples processed by conventional sintering. The highest hardness (~475 HV) and compressive strength (~450 MPa) are that of the composite containing 5 wt.% HfB_2_ associated with a porosity reduction of approximately 10%. These improvements are attributed to the enhanced densification and microstructural refinement achieved via microwave processing. The findings underscore the potential of HfB_2_ addition and microwave sintering in tailoring the structure–property relationships of the Ti_2_AlC composites, enabling applications at high-temperature filtration, thermal barriers, and self-lubricating components.

## 1. Introduction

MAX phases are represented by the convention formula Mn+1AXn, which are included in the family of ternary layered carbides and nitrides. These exhibit an outstanding combination of metallic and ceramic properties. Ti_2_AlC is a member of the 211 MAX phase group that has gained significant attention for its high stiffness (elastic modulus of ~277 GPa), moderate hardness, good thermal and electrical conductivity, and notable damage tolerance [1,2,3]. However, its application remains limited in high-load or harsh thermal environments because of its insufficient hardness and strength [4]. Thus, tailoring its microstructure through reinforcement and advanced processing routes is essential to fully exploit its potential. This combination of properties makes the Ti_2_AlC suitable for high-temperature structural applications in aerospace, automotive, and nuclear industries [5,6]. However, despite these advantages, the Ti_2_AlC exhibits moderate mechanical strength and hardness, which limits its broader use in load-bearing applications. Therefore, reinforcing Ti_2_AlC with secondary phases such as SiC and Al_2_O_3_ has been explored to improve its mechanical properties [7,8,9].

Some investigations have explored the versatility of the Ti_2_AlC MAX phase ceramics in a variety of functional applications. Liu et al. [10] have enhanced microwave absorption via Lewis acid molten salt etching of the Ti_2_AlC powders, while Suh et al. [11] have demonstrated the thermal self-healing capability of the phase at different temperatures. Laska et al. [12] have applied the Ti_2_AlC coatings on γ-TiAl alloys, aiming to improve oxidation resistance. Other studies have explored spark plasma sintering [13], the MAX metal lamellar structures with good mechanical properties [14], and corrosion resistance in saline solution [15]. Furthermore, Huang et al. [16] have discussed the thermal stability of Ti_2_AlC and Ag/Ti_2_AlC composites.

Among various ceramic reinforcements, the HfB_2_ stands out for its exceptional hardness (~25 GPa), high melting point (>3000 °C), and excellent thermal stability. Its combination with the Ti_2_AlC has a potential to create composites suitable for extreme environments. One of the most effective reinforcement strategies involves the introduction of high-strength ceramic phases. For instance, adding Al_2_O_3_ fibers to the Ti_2_AlC composites has resulted in enhanced flexural strength (~697 MPa) and fracture toughness (~9.83 MPa·m^1/2^), comparable to SiCf/SiC composites [8]. Similarly, reinforcing the Ti_2_AlC with TiB_2_ through in situ reaction hot pressing improved its compressive strength and oxidation resistance at high temperatures (600–900 °C) [17]. Nanostructured reinforcements, such as the nano-Ti_2_AlC particles, have also shown remarkable improvements in the mechanical response of the Ti_2_AlC-based composites [18,19]. The introduction of submicron-sized Ti_2_AlC particles into TiAl matrices increased compressive strength to ~2171 MPa and fracture strain to ~31% due to enhanced interfacial bonding and controlled dislocation mobility [19]. Moreover, nano-Ti_2_AlC reinforcements into the TiAl matrices have increased the strength–plasticity synergy by inducing dislocation multiplication and twin formation at the grain boundaries. This mechanism improves the load transfer and promotes energy dissipation during deformation. This has promoted composites with higher toughness and damage tolerance [19].

The incorporation of Ti_2_AlC into composite materials has increased both the mechanical and thermal properties, which makes the potential candidate suitable for high-temperature applications. Studies have shown that the Ti_2_AlC significantly improves the oxidation resistance and mechanical strength of ultrahigh-temperature ceramics, particularly in the HfB_2_-based systems, by reducing porosity and increasing densification during sintering [20]. Furthermore, the addition of ceramic reinforcements, e.g., TiB_2_ and Al_2_O_3_, into the Ti-Al-C matrices has been reported [21,22]. Consequently, its corresponding fracture toughness and wear resistance are increased, which is associated with superior strength and thermal stability [21,22]. Nevertheless, only a limited number of studies have explored the effect of the HfB_2_ content on the Ti_2_AlC, especially in combination with alternative sintering routes such as microwave processing. This knowledge gap motivates the current investigation. The formation of the core–shell structures, such as ZrC/Ti_2_AlC, further refines the microstructure, leading to improved hardness and oxidation resistance under extreme conditions [23]. These findings suggest that optimizing the composition and processing conditions of the Ti_2_AlC-based composites provides the development of advanced materials for aerospace, automotive, and energy applications.

Microwave sintering has emerged as a promise-processing route for Ti_2_AlC-based composites [24,25,26,27,28,29,30]. Unlike conventional sintering, where heat is transferred from the surface to the interior, microwave sintering generates heat uniformly throughout the sample, reducing thermal gradients and promoting rapid densification [31,32]. This leads to finer microstructures, reduced porosity, and improved mechanical properties.

This study aims to investigate the influence of the different HfB_2_ contents (i.e., 0, 2.5, 5, 7.5, and 10 wt.%) on the microstructure and mechanical properties of the Ti_2_AlC composites processed through both the conventional and microwave sintering. The goal is to identify the optimal HfB_2_ content and sintering route. With this, improved hardness and compressive strength are attained while the porosity is decreased. The findings provide valuable insights into the development of high-performance Ti_2_AlC-based composites for structural and high-temperature applications. The contribution of this study lies in the correlation between the microstructural array and porosity, consequently affecting the mechanical performance under two distinct sintering regimes positively. With this, resulting insights for future applications in high-temperature filtration, wear-resistant components, and energy systems are obtained.

## 2. Materials and Methods

### 2.1. Composites Preparation and Powder Metallurgy Stages

The initial powders of Ti (99.4%, 100 mesh), Al (99.8%, 325 mesh), and TiC (99%, 325 mesh) were used to constitute the Ti_2_AlC (TAC) composition, following a stoichiometric Ti:Al:TiC ratio. The powders were mixed manually in an agate mortar for approximately 20 min at room temperature (25 ± 2 °C), without binder or lubricant, to ensure homogeneity and to avoid particle size modification. No ball milling was utilized for this reason. For the sample compaction, a cylindrical steel die (ASTM A6 [33], diameter 8 ± 0.5 mm) was used. A uniaxial pressure of 975 ± 50 MPa was applied for 2 min. The green compacts were then sealed in quartz ampoules under vacuum (10^−2^ mbar). The heat treatments were performed at 400 °C, 600 °C, and 1200 °C ± 10 °C for 30 min, using both the conventional and microwave sintering systems (FMO1600, Fortelab, São Carlos, Brazil). For the microwave process, a frequency of 2.45 GHz and a maximum power of 1.2 kW were applied. Temperature during microwave sintering was monitored by using a thermocouple (Type K) and maintained within a precision of ±10 °C throughout the treatment. The sintering parameters of 1200 °C for 30 min, experimentally optimized, were chosen to balance the densification of the Ti_2_AlC matrix with the preservation of the HfB_2_ reinforcement, preventing excessive grain growth and interfacial reactions. Microwave sintering differs from conventional heating due to its rapid volumetric heating and strong interfacial field interactions. This intensifies local diffusion and densification kinetics at particle contacts, even with short holding times. Consequently, the short cycle and low thermal budget limit long-range diffusion, preserving the distribution and integrity of the HfB_2_ phase. Triplicate is adopted for each condition, and the resulting property values (e.g., hardness, porosity) exhibited short standard deviations (<10%). This suggests that possible temperature fluctuations have no substantial effects on the reproducibility and reliability of the results. The literature reports that such slight fluctuations have not significantly affected the densification or phase formation in the MAX-phase ceramics processed by microwave sintering. After synthesizing the Ti_2_AlC phase, HfB_2_ powder (99%, 325 mesh, Sigma-Aldrich) was added to distinctive contents (0, 5, 7.5, and 10 wt.%). The HfB_2_ powder was commercially purchased from Sigma-Aldrich (St. Louis, MO, USA), with a nominal purity of 99% and an average particle size corresponding to 325 mesh (~ 45 μm). The as-received powders are constituted of irregular-shaped particles, as provided by the supplier. It must be noted that no additional purification or milling was used. The intention behind this was to avoid altering its original particle size distribution. The HfB_2_ and Ti_2_AlC powders were manually homogenized, as aforementioned. Sequentially, the resulting mixtures were compacted, encapsulated, and heat-treated under the same conditions. Before the sintering, each one of the examined samples was adequately weighed (1.2 ± 0.1 g). Three specimens of each composition were adopted to ensure statistical reliability.

### 2.2. Microstructural Observations

X-ray diffraction (XRD) analysis was performed using a diffractometer (Panalytical diffractometer, X’Pert model, Malvern, UK) in Bragg–Brentano geometry, equipped with CuKα radiation and a Ni filter. Scanning electron microscopy (SEM TESCAN^®^ model VEGA3, Brno, Czech Republic) images were acquired. An energy-dispersive X-ray spectroscopy (EDS) detector associated with SEM analyses was also utilized. For SEM characterizations, all samples were polished using SiC papers (up to 1200 grit) and polished with 1 μm diamond paste. SEM observations were performed at an accelerating voltage of 15 kV and a working distance of ~10 mm.

### 2.3. Relative Density, Hardness, and Compressive Strength Measurements

The Archimedes method was used to determine both the green and sintered densities. For this purpose, a specific gravity measurement kit SMK-401 (Shimadzu, Kyoto, Japan) was used. Vickers hardness measurements (Buehler—Wilson VH1102, Lake Bluff, IL, USA), using 10 s preloading, were attained on the polished surface. At least 10 indentations were carried out for each examined sample. The measurements were taken on polished surfaces to minimize the influence of surface irregularities, and average values were calculated.

The mechanical characterization was carried out using a universal electro-hydraulic servo testing machine operating under controlled ambient conditions (25 ± 2 °C). The tests were conducted at a constant crosshead speed of approximately 25 mm/min, corresponding to a nominal strain rate of 2 × 10^−4^ s^−1^, in accordance with standard procedures for quasi-static compression testing. This setup ensures consistent loading conditions for the accurate assessment of the mechanical performance. The tests were conducted according to ASTM standard E8/E8M-24 [34], and triplicate experimentation was adopted to ensure reproducibility. The cylindrical specimens with diameters of 8 (±0.5) mm and a height-to-diameter ratio of ~1.2 were prepared and examined. No lubricant was applied, and the temperature at 25 (±2) °C was also adopted.

## 3. Results

Figure 1 shows a typical scanning electron microscopy micrograph of the heat-treated Ti_2_AlC/5HfB_2_ composite at 1200 °C for 30 min. This micrograph reveals a typical morphology obtained when a conventional powder metallurgy route is applied. Both the Ti_2_AlC and HfB_2_ grains are clearly identified.

Energy-dispersive spectroscopy (EDS) analysis has revealed the elemental distribution, as shown in Figure 1. As expected, Ti and Al are the dominant elements. It should be noted that C quantification is not precise due to low spectral energy. Figure 1 also shows a more uniform distribution of the HfB_2_ spheroid-like particles, which appears closely related to the observed porosity level. This seems to be intimately associated with the porosity level attained. The absence of interfacial microcracking suggests good compatibility between the Ti_2_AlC matrix and HfB_2_ reinforcement.

A homogeneously distributed porous morphology is observed, which is consistent with the apparent porosity results shown in Figure 2. The porosity reaches approximately 18 ± 1.3%.

Figure 2a shows the XRD patterns of the Ti2AlC/xHfB_2_ composites. The samples were conventionally sintered in a furnace at 1200 °C for 30 min. The XRD patterns reveal the Ti_2_AlC formation in all examined samples. This phase belongs to space group 194, with lattice parameters a = 0.304 nm and c = 1.374 nm. Additionally, the HfB_2_ is identified in space group 191, with the lattice parameters a = 0.314 nm and c = 0.347 nm [35]. The crystallite size and residual microstrain of the Ti_2_AlC sample were evaluated by using X-ray diffraction peak broadening analysis. Considering those four most intense diffraction peaks, the average crystallite size was found to be approximately 55.7 nm, while the average microstrain was about 0.11%. These values are characteristics of the well-crystallized MAX phase ceramics, which indicates that efficient sintering is promoted [36,37]. The relatively large crystallite size suggests substantial grain growth during heat treatment, which is consistent with a typical sample that was conventionally sintered for a long-term period at high temperatures. Meanwhile, the moderate level of residual microstrain indicates a good degree of structural relaxation, implying that most of the internal stresses generated during synthesis were successfully decreased. These findings are in good agreement with previously reported investigations when conventional sintering of the Ti_2_AlC is applied [36,37].

Although the HfB_2_ content increases, no deleterious effect on the Ti_2_AlC phase formation kinetics was observed, which is expected since the HfB_2_ has a low self-diffusion coefficient. The intensity of the HfB_2_ phase increases with the increase in the HfB_2_ content. No peak shift was observed for the Ti_2_AlC or HfB_2_ reflections, suggesting minimal lattice distortion due to the additive.

When sintering is carried out at 1200 °C for 30 min, the formation of the TiC as an intermediate and thermodynamically stable phase is observed. This stability is consistent with previous studies [38,39,40] on the Ti–Al–C system, which demonstrates that the TiC tends to persist at high temperatures due to its high melting point and strong crystallography associated with the Ti_2_AlC. The presence of the TiC at this stage is commonly attributed to local stoichiometric variations or incomplete reactions, as also previously reported [38,39,40]. This phase is identified at 2θ angles of approximately 42° and 36°, corresponding to planes (111) and (200), respectively, as also previously reported [35].

The apparent densities were determined using the Ti_2_AlC theoretical density and the rule of mixtures. The density values, wet mass (*m_wet_*), and dry mass (*m_dry_*), along with the apparent porosity (AP) of the material, were determined as described in Equation (1).(1)$$AP\;\left(\%\right)=\frac{m_{wet\;}-m_{dry}}{m_{wet}-m_{app}}\times \;100$$
where *m_app_* represents the apparent mass of the examined samples.

The lowest porosity level is that of the Ti_2_AlC/5HfB_2_ composite containing 5 wt.% HfB_2_. When comparing the composite without the HfB_2_ and the Ti_2_AlC/5HfB_2_ composite, a decrease of approximately 6% is observed. The microhardness values of each examined composite are correlated with the porosity results, as shown in Figure 2b.

An inverse correlation between the hardness and porosity as a function of the HfB_2_ content was established. The highest hardness is that of the Ti_2_AlC/5HfB_2_ composite (375 ± 87 HV), representing an increase of approximately 50% in HV values. Additionally, this composite sample also exhibited the lowest porosity level. This trend confirms that densification plays a predominant role in determining the mechanical response over potential grain refinement or phase changes.

The composition with 5% HfB_2_ exhibits the lowest porosity and highest hardness due to an optimal balance between sintering efficiency and mechanical reinforcement. At this concentration, the HfB_2_ particles are uniformly distributed, promoting better grain cohesion and reducing porosity through enhanced mass diffusion. The presence of the HfB_2_ also strengthens the matrix by impeding dislocation movement, leading to higher hardness. High HfB_2_ contents induce agglomeration and the TiC formation, increasing porosity and reducing mechanical performance. Therefore, 5% HfB_2_ represents the ideal balance between structural integrity and mechanical strength.

Figure 2c displays the stress (σ) vs. strain (ε) curves in engineering format for the Ti_2_AlC/xHfB_2_ composites heat-treated at 1200 °C for 30 min. These results are consistent with the previously mentioned hardness and porosity values. The highest ultimate compressive strength (UCS) of ~345 MPa is that of the Ti_2_AlC/5HfB_2_ composite, while the intermediate UCS (~225 MPa) corresponds to the Ti_2_AlC/7.5HfB_2_ composite. The lowest UCS (~153 MPa) is that of the composite without the HfB_2_, demonstrating the strengthening effect of this dopant up to 5 wt.%.

This trend is also evident in the experimental yield strengths (YSs), which are ~250, 160, and 105 MPa, respectively. These YS values are approximately 1.4 times lower than the corresponding UCS values. From a mechanical perspective, these results appear to be associated with the inhibition of cavity formation and intergranular sliding. At room temperature (25 ± 2 °C), the plastic deformation induces the kinking and delamination of the Ti_2_AlC grains. Since both the Ti_2_AlC/2HfB_2_ and Ti_2_AlC/10HfB_2_ composites exhibit lower UCS than the other examined samples, their corresponding curves were not considered in the discussion. Standard deviation values remained below 10% in all cases, indicating reliable mechanical behavior and reasonable sample reproducibility.

The microwave heat treatment is recognized as a distinctive manufacturing route for sintering the Ti_2_AlC/xHfB_2_ composites. This process involves emitting microwaves into the material, which absorbs the electromagnetic energy and generates heat [41]. When conventional sintering is adopted, only the material is heated, and heat is not transferred among objects by conduction, radiation, or convection. In contrast, during the microwave heat treatment, the heat is initially generated inside the material and then distributed throughout the entire volume [42]. Zhou et al. have also used microwave heating to sinter the Ti_2_AlC, achieving 96.6% phase purity [43]. Using the same processing parameters, new samples were produced and subjected to the microwave heat treatment at 1200 °C for 30 min.

Figure 3 shows typical SEM images of the sintered Ti_2_AlC/5HfB_2_ composite subjected to the microwave heat treatment at 1200 °C for 30 min. A typical BSE image shows a polished surface with a different microstructural arrangement compared to the conventionally treated sample. A microstructure with more needle-like particles is observed. Although needle-like particle clusters are typically characterized in the M_2_AX alloys [43,44,45], the smallest particles in the clusters are observed when microwave sintering is adopted, as depicted in Figure 3. This morphology, more prevalent under microwave sintering, is associated with rapid diffusion and with a directional growth mechanism triggered by internal heating.

From the XRD results, as demonstrated in Figure 4a, it is worth noting that a complete formation of the Ti_2_AlC phase is attained. In addition, it is also found that the intensity peaks (at XRD patterns) of the HfB_2_ are increased with the increasing HfB_2_ content. This occurrence is also observed when the conventional heat treatment is also carried out. Also, the intermetallic TiC as a secondary phase is constituted.

For the Ti_2_AlC sample processed by microwave-assisted sintering, the analysis of the four most intense X-ray diffraction peaks revealed an average crystallite size of approximately 38.1 nm and an average residual microstrain of 0.10%. The crystallite size is smaller than the conventionally sintered sample (55.7 nm). This demonstrates the effectiveness of microwave processing in limiting grain growth due to its rapid heating rates and shorter dwell times. The quantitative phase analysis highlights the effectiveness of both sintering methods in promoting the formation of the Ti_2_AlC phase. However, conventional sintering led to a slightly higher phase purity (97 vol%) compared to microwave sintering (92 vol%). This difference may be attributed to the rapid heating rates and potential local thermal gradients inherent to microwave processing, which can favor the persistence of secondary phases such as TiC. Despite the marginally lower Ti_2_AlC content in the microwave-sintered samples, the method still offers significant advantages in terms of energy efficiency and microstructural refinement.

Figure 4b shows both the apparent porosity and hardness results of the Ti_2_AlC/xHfB_2_ composites with the three distinct HfB_2_ contents. Consistent with our previous observations using conventional heating, the Ti_2_AlC/5HfB_2_ composite exhibits a lower porosity level. When the 5 wt.% HfB_2_ content is used, the porosity level decreases by about 10% (i.e., from ~23% to ~13%). In contrast, when the 7.5 wt.% HfB_2_ content is used, the porosity increases, as also observed with the conventional sintering. The decrease in the porosity is associated with an increase in the hardness. The highest UCS value is that of the Ti_2_AlC/5HfB_2_ composite (~474 ± 33 HV), followed by the composites with 7.5 and 2.5 wt.% HfB_2_, respectively.

An analogous pattern emerges under the conventional sintering. These results indicate that the microwave sintering positively influences the mechanical behavior of the Ti_2_AlC/5HfB_2_ composite.

Figure 4c shows the experimental stress (σ) vs. strain (ε) curves, in engineering format, of the heat-treated Ti_2_AlC/xHfB_2_ composites at 1200 °C for 30 min at environmental temperature (25 ± 2 °C). As expected, these were similar to the results obtained with the conventional heat treatment. When the HfB_2_ content is increased up to 5 wt.%, the compressive strength is increased. The lowest UCS value (~252 ± 27 MPa) and YS of ~180 MPa are observed in the Ti_2_AlC/0HfB_2_ sample (without HfB2). The highest UCS (~450 ± 35 MPa) is that of the sample with 5 wt.% HfB_2_, while the intermediate UCS (~327 ± 27 MPa) was obtained in the Ti_2_AlC/7.5HfB_2_ composite. Both the Ti_2_AlC/5HfB_2_ and Ti_2_AlC/7.5HfB_2_ composites exhibit YS values approximately 1.4 times lower than their corresponding UCS values. This is consistent with results when the conventional heat treatment is adopted. Due to the microwave treatment and the resulting finer morphology, oscillations in the stress–strain curve, mainly after the elastic region, are observed. This seems to be associated with the number of dislocations generated when the needle-like morphology is mechanically demanded.

When comparing the mechanical responses of the conventional and microwave treatments, the UCS and YS values of the former are approximately 1.6 times higher than the latter ones. This trend is consistent not only for the 5 wt.% HfB_2_ content but also for all other examined composites. When comparing the two treatments (Figure 2 and Figure 4), an interesting observation regarding elongation is noted. For instance, the Ti_2_AlC composites without the HfB_2_ content exhibit average elongations of ~4%, regardless of the heat treatment adopted. When considering the HfB_2_ content, the conventional treatment results have demonstrated similar elongations.

In contrast, the microwave treatment results promote slightly decreased elongation values. However, this decrease is more pronounced in the 7.5 wt.% HfB_2_ composite (~3.5% vs. ~6% with conventional treatment). Although the elongation typically decreases with increasing UCS, a more pronounced deleterious effect is observed in composites with higher HfB_2_ content.

## 4. Discussion

A comparison between conventional sintering and microwave-assisted sintering reveals significant microstructural implications (Figure 5), with a direct impact on the final properties of the materials. While conventional heating relies on external thermal conduction, promoting gentler temperature gradients and prolonged processing times, microwave sintering operates through direct and selective volumetric heating. This intensifies the diffusion kinetics at localized scales. This characteristic results in the formation of more refined microstructures (Figure 5a,b). Associated with this, a noticeable reduction in the average particle size and residual porosity is achieved. The direct interaction of the electromagnetic field with the ceramic matrix favors the non-thermal sintering mechanisms, such as intergranular coupling and differential heating. These effects contribute to more homogeneous densification and improved control of the grain morphology. The powder mixture was manually homogenized in an agate mortar for approximately 20 min at room temperature (25 ± 2 °C), without the addition of binders or lubricants, to preserve the original particle size and morphology. Although no SEM or EDS characterization was conducted on the mixed powder, the post-sintering microstructural analysis at low magnification indicates a satisfactory distribution of HfB_2_ within the Ti_2_AlC matrix, suggesting that the mixing protocol was effective in achieving initial homogeneity.

These mentioned microstructural modifications not only affect the mechanical behavior, inducing improvements in both the hardness and toughness, but they also indicate a promising path for the development of advanced ceramics. This suggests superior performance with applications at high temperatures. These findings highlight the potential of microwave sintering as a disruptive technology for the engineering of refractory materials and functional composites.

Microwave heating has promoted more efficient interparticle cohesion and refinement of the microstructure. This has also decreased the apparent porosity. The quantitative analyses reveal that the sample sintered by using the microwave exhibited a porosity of 13%, in contrast to the 18% observed when the sample was subjected to conventional sintering (Figure 6). This means a relative reduction of approximately 28%, highlighting the superior effectiveness of microwave heating in eliminating intergranular pores. Such performance is associated with the volumetric and rapid nature of microwave heating, which promotes uniform densification and accelerates the diffusion mechanisms within the material, even in shorter treatment times.

A comparative analysis of the mechanical properties of the examined composites under two different heat treatments is shown in Figure 6. This analysis indicates that lower porosity levels improve mechanical performance. Furthermore, the morphology refinement is achieved with the microwave treatment.

From the experimental results of the mechanical strength and densification levels, another important parameter is determined, namely the specific strength (SS), which is defined as the UCS divided by the sintered density. This reflects the combination of the mechanical strength and lightweight effect. The specific strength (SS) for the MW (microwave) treatment is ~118 ± 6 × 10^3^ m^2^/s^2^ (ρ = 3.64 × 10^3^ kg/m^3^), while the CV (conventional) treatment yields a SS of ~95 ± 5 × 10^3^ m^2^/s^2^ (ρ = 3.46 × 10^3^ kg/m^3^) (Table 1). These results suggest that the MW treatment is a viable alternative for the sintering of the studied composites. Additionally, microwave sintering produces the desired phase while preventing TiC formation typically constituted during conventional sintering. This mechanical improvement, combined with the suppression of the undesirable TiC formation, positions microwave sintering as a promising method for the cost-effective production of Ti_2_AlC-based components.

Barsoum et al. [1] have reported that the compressive strength varies with the grain size. Specifically, a coarser grain size (100 to 200 μm) results in a compressive strength of ~390 MPa, whereas a finer grain size (~25 μm) leads to a compressive strength of ~540 MPa.

In this investigation, the starting particle sizes were ~150 μm (100 mesh) for Ti and 45 μm (325 mesh) for both Al and graphite. These results are consistent with those previously reported by Barsoum et al. [1]. However, while Barsoum’s results were obtained with conventional processing, the current study demonstrates that similar or superior properties can be achieved using microwave sintering, potentially reducing energy costs and time. Additionally, the findings suggest that the Ti_2_AlC phase can be manufactured using powder metallurgy without additional processing steps, resulting in lower production costs.

This study demonstrates that the mechanical performance of the Ti_2_AlC/xHfB_2_ composites is significantly influenced by the heat treatment method and the HfB_2_ content. The results indicate that the microwave heat treatment effectively improves both the densification and mechanical strength of the composites due to enhanced microstructural refinement and reduced porosity. The highest ultimate compressive strength (UCS) of ~450 MPa and the highest specific strength (SS) of ~118 × 10^3^ m^2^/s^2^ are obtained for the composite with 5 wt.% HfB_2_ under microwave treatment. It is important to note that microwave sintering operates via a fundamentally different mechanism compared to conventional heating, promoting rapid volumetric heating and strong interfacial field interactions. This leads to enhanced local diffusion and densification kinetics, especially at particle contacts, even under short dwell times. However, due to the lower overall thermal budget and short sintering cycle, microwave sintering naturally limits long-range diffusion, which in turn preserves the distribution and integrity of the HfB_2_ phase within the matrix [30].

The observed increase in hardness may be partially attributed to the presence of HfB_2_ particles, which, according to previous studies, can act as obstacles to dislocation motion in ceramic-reinforced composites [19,46,47]. While this mechanism is consistent with the literature, it is important to note that it was not directly verified in this work due to the absence of TEM or EBSD analysis.

In contrast, the conventional sintering resulted in a higher porosity (~18%) and a lower specific strength (~95 × 10^3^ m^2^/s^2^). The microwave treatment has promoted the formation of the Ti_2_AlCphase while effectively suppressing the formation of the TiC. This is more prominent in the conventionally treated samples. Furthermore, the increase in the HfB_2_ content beyond 5 wt.% led to an increase in the porosity and a decrease in mechanical strength, highlighting the importance of optimizing the HfB_2_ content.

The findings confirm that microwave sintering is a viable alternative to produce Ti_2_AlC-based composites with improved mechanical properties and reduced production costs. The enhanced mechanical behavior and microstructural stability achieved with microwave sintering indicate that this method is a promising approach for manufacturing high-performance Ti_2_AlC composites. Future work should focus on exploring the long-term mechanical and thermal stability of these composites under different environmental conditions. In summary, this study provides a pathway for engineering Ti_2_AlC-based composites with superior performance using scalable and energy-efficient techniques. This advances the practical implementation of MAX phase ceramics at high temperatures and in high-stress environments.

## 5. Conclusions

Based on the experimental results, the following conclusions can be drawn:Both the conventional and microwave-assisted sintering enables the successful formation of the Ti_2_AlC phase using an optimized Ti:TiC:Al stoichiometry (1.2:1:1), achieving phase purity of up to 97%.Incorporating 5 wt.% HfB_2_ maximizes mechanical performance, with substantial improvements in hardness, compressive strength, and reduced porosity, independent of the sintering route adopted.The microwave processing further refines the microstructural arrays, resulting in the mechanical properties approximately 1.6 times higher than those obtained by the conventional methods.These findings establish powder metallurgy with microwave sintering as an efficient, scalable route for the advanced Ti_2_AlC-based composites suitable for demanding high-temperature applications.

## Figures and Tables

**Figure 1 materials-18-02693-f001:**
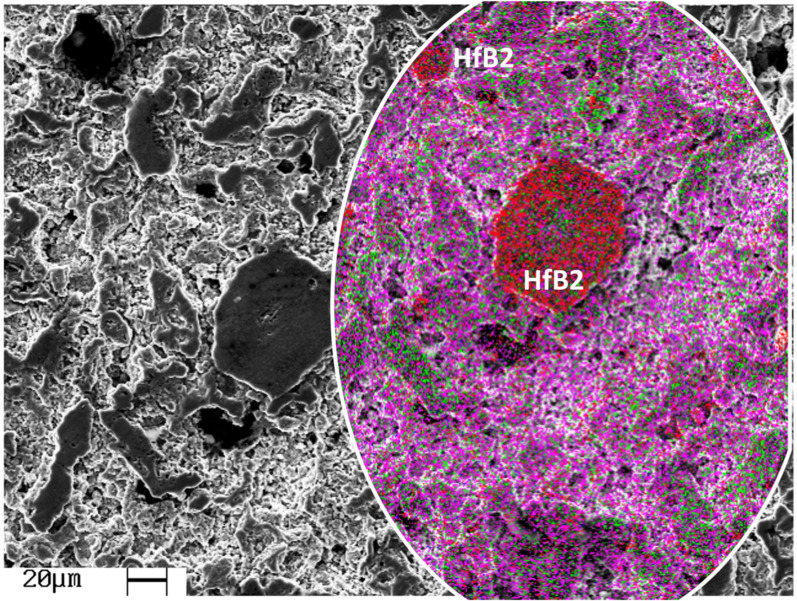
Typical SEM images of the heat-treated Ti_2_AlC/5.0HfB_2_ composites at 1200 °C for 30 min. The inset presents the EDS elemental mapping, where titanium (Ti) is represented in magenta, aluminum (Al) in green, and hafnium (Hf) in red. The elemental maps highlight the spatial distribution of the constituents and support the identification of the HfB_2_ reinforcement within the Ti_2_AlC matrix.

**Figure 2 materials-18-02693-f002:**
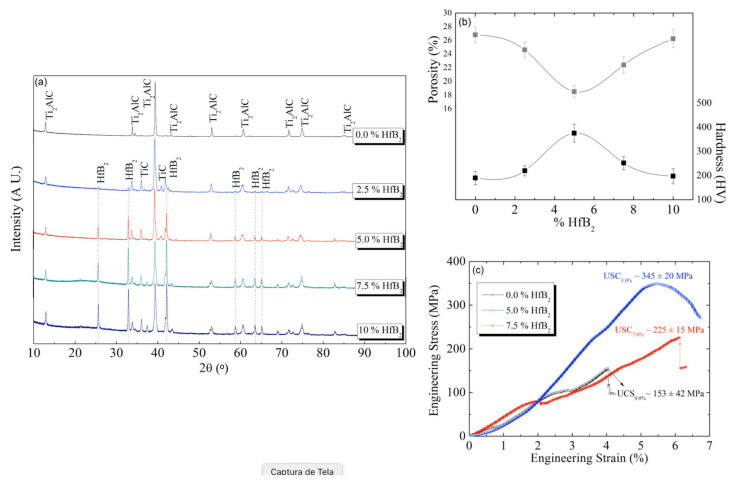
(**a**) XRD results of the Ti_2_AlC/xHfB_2_ conventionally sintered at 1200 °C for 30 min. The ‘x’ index corresponds with the distinctive HfB_2_ contents; (**b**) experimental results of the apparent porosity (upper) and hardness (lower) of the heat-treated Ti_2_AlC/xHfB_2_ composites conventionally at 1200 °C for 30 min; (**c**) experimental curves of the stress vs. strain of the conventionally heat-treated Ti_2_AlC/xHfB_2_ composites at 1200 °C for 30 min.

**Figure 3 materials-18-02693-f003:**
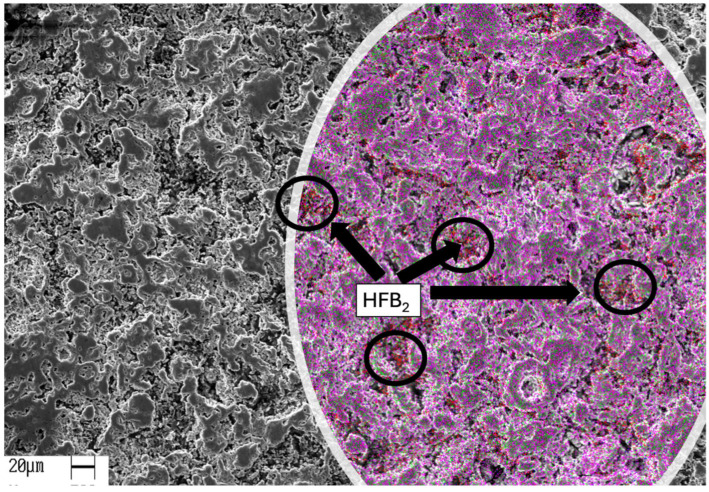
Typical SEM images of the heat-treated Ti_2_AlC/5.0HfB_2_ composites at 1200 °C for 30 min using microwave heating. The inset presents the EDS elemental mapping, where titanium (Ti) is represented in magenta, aluminum (Al) in green, and hafnium (Hf) in red. The elemental maps highlight the spatial distribution of the constituent elements and support the identification of the HfB_2_ reinforcement within the Ti_2_AlC matrix.

**Figure 4 materials-18-02693-f004:**
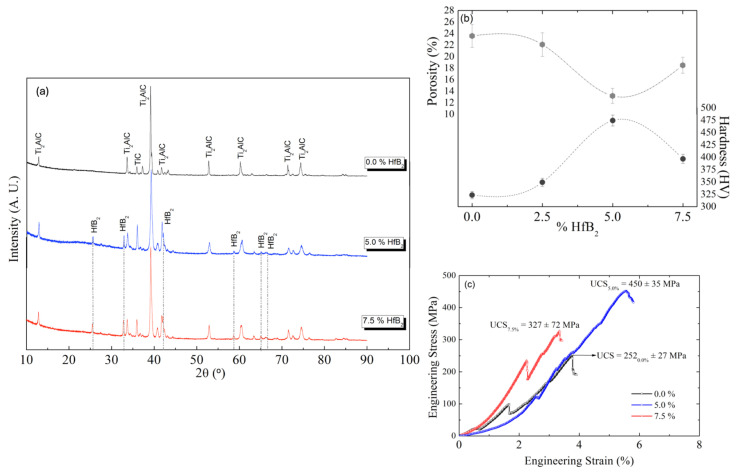
(**a**) XRD results of the heat-treated Ti_2_AlC/0HfB_2_, Ti_2_AlC/5HfB_2_, and Ti_2_AlC/7.5HfB_2_ composites by using the microwave treatment at 1200 °C for 30 min. (**b**) Apparent porosity (upper) and hardness (lower) of the heat-treated Ti_2_AlC/xHfB_2_ composites using the microwave treatment at 1200 °C for 30 min. (**c**) Stress vs. strain engineering curve of the heat-treated Ti_2_AlC/xHfB_2_ composites using the microwave (MW) at 1200 °C for 30 min.

**Figure 5 materials-18-02693-f005:**
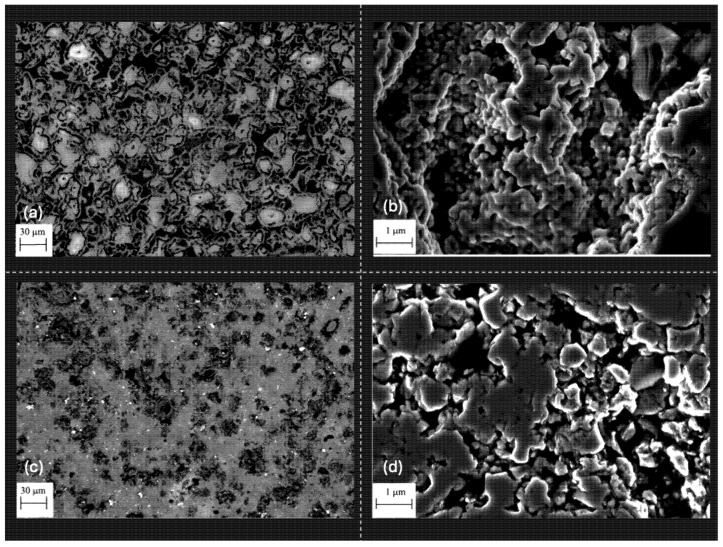
SEM micrographs of the Ti_2_AlC composite with 5 wt.% HfB_2_ sintered via (**a**,**b**) microwave-assisted heating and (**c**,**d**) conventional sintering. The microstructure obtained by the microwave sintering shows densification and refined grain distribution improved when compared with the conventionally sintered sample.

**Figure 6 materials-18-02693-f006:**
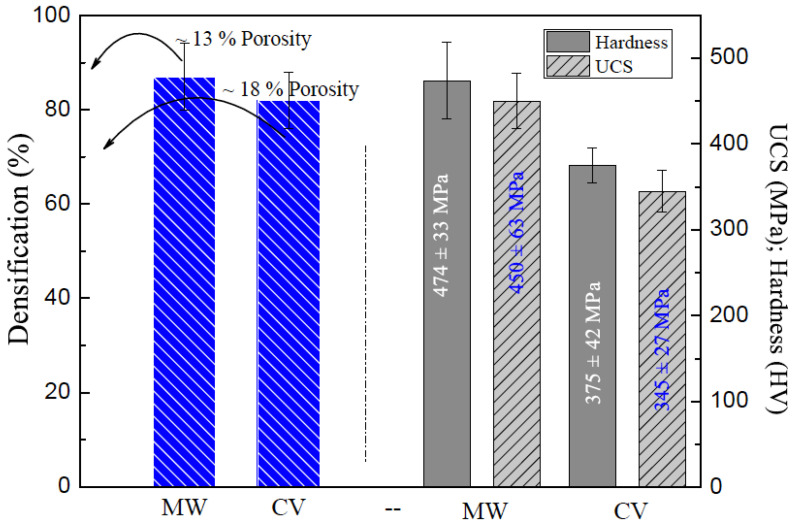
Comparative analysis concatenating mechanical behavior (UCS and hardness) and densification of the Ti_2_AlC/xHfB_2_ composites using both the MW (microwave) and CV (conventional) treatments.

**Table 1 materials-18-02693-t001:** Comparative values of the physical and mechanical properties of the samples sintered by conventional and microwave heating. The measurements of apparent porosity, Vickers hardness, compressive strength limit, and specific resistance are compiled, highlighting the distinct effects of sintering mechanisms on the microstructure and mechanical performance of the material.

Properties	Sintering Method	
	Conventional (CV)	Microwave (MW)
Porosity (%)	18 ± 1.3	13 ± 1.1
Hardness (HV)	375 ± 42	474 ± 33
Ultimate Strength (MPa)	345 ± 27	450 ± 35
Specific Strength (10^3^ m^2^/s^2^)	95 ± 5	118 ± 6
a lattice parameter (nm)	0.304 ± 0.01	0.3043 ± 0.01
c lattice parameter (nm)	1.374 ± 0.001	1.363 ± 0.001
Crystallite size (nm)	55.7 ± 8	38.1 ± 5
Lattice strain (%)	0.11 ± 0.02	0.10 ± 0.02

## Data Availability

The raw data supporting the conclusions of this article will be made available by the authors on request.

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
