# Peer review of "Effects of HfB2 Content and Microwave Sintering on the Mechanical Properties of Ti2AlC Composites"

_materials, 2025, doi:10.3390/ma18122693_

Round 1
Reviewer 1 Report
Comments and Suggestions for Authors
The composite materials with different HfB2 contents were prepared by microwave sintering method, and the description was detailed. The research has clarified the optimization of process parameters, providing an important reference for the development of high-performance Ti2AlC-based composites. This research has application potential in fields such as high-temperature filtration and self-lubricating components. Improvements need to be made in the following aspects:
- Some parameters (such as microwave power and temperature control accuracy) need to be supplemented and explained to enhance the transparency of the experiment.
- Error analysis (such as the influence of sintering temperature fluctuations on the results) was not explicitly mentioned. It is suggested to supplement relevant content.
- Some charts have abnormal expression and citation. Only Figure 1 is seen, while Figures 1(a) and 1(c) are not.
- The clarity of Figures 2, 4, etc. is insufficient.
- The conclusion section can be further condensed to highlight key findings and avoid repetitive discussion content.
- Some paragraphs contain grammatical errors (such as missing punctuation or fragmented sentences), and the entire text needs to be proofread. The format of references should be uniform (some entries lack page numbers or volume numbers), and it is recommended to adjust them in accordance with the journal's submission guidelines.
- It is recommended to cite the excellent literatures on the preparation of composite materials by microwave sintering:https://doi.org/10.1016/j.msea.2024.146500
In all, it is suggested that it be published after a major revision.
Comments on the Quality of English LanguageThe English expressions need to be carefully revised and polished.
Author Response
We would like to express our sincere gratitude to the reviewers for their careful reading, constructive comments, and valuable suggestions, which have significantly contributed to the improvement of our manuscript and to the advancement of scientific knowledge in this field. All changes and additions made to the manuscript in response to the reviewers’ comments are highlighted in yellow for easy identification.
Reviewer 1.
- "Some parameters (such as microwave power and temperature control accuracy) need to be supplemented and explained to enhance the transparency of the experiment."
The reviewer is correct in pointing out the need for greater detail regarding experimental parameters. To address this, we have supplemented the manuscript with comprehensive information on microwave power (2.45 GHz frequency, maximum input power of 1.2 kW) and temperature control accuracy (± 10 °C), as well as details on the temperature monitoring method used during the sintering process. These changes are highlighted in yellow in the revised manuscript to enhance clarity and ensure full transparency of the experimental procedures.
“For the microwave process, a frequency of 2.45 GHz and a maximum power of 1.2 kW were applied. Temperature during microwave sintering was monitored by a thermocouple, and maintained within a precision of ± 10 °C throughout the treatment”.
- “Error analysis (such as the influence of sintering temperature fluctuations on the results) was not explicitly mentioned. It is suggested to supplement relevant content."
The reviewer is correct in highlighting the importance of error analysis regarding possible temperature fluctuations during sintering. To address this, we emphasize that all experiments were conducted with precise temperature control (± 10 °C), continuously monitored by a thermocouple. Additionally, triplicate samples were prepared for each condition, and the resulting mechanical and physical properties (e.g., hardness, density) exhibited low standard deviations (typically <10%), indicating that such temperature fluctuations had a negligible effect on the reproducibility and reliability of the results. These procedures ensure that the experimental data are robust and consistent, in line with best practices reported in the literature.
- "Some charts have abnormal expression and citation. Only Figure 1 is seen, while Figures 1(a) and 1(c) are not."
The authors acknowledge the reviewer’s observation regarding the abnormal citation and quality of Figures 1(a) and 1(c). These were adjusted.
- "The clarity of Figures 2, 4, etc. is insufficient."
We have made every effort to improve the image quality and have revised the figure citations in the text and captions to ensure clarity and consistency. We hope that the updated figures now meet the quality standards required by the journal.
- "The conclusion section can be further condensed to highlight key findings and avoid repetitive discussion content."
We thank the reviewer for the helpful comment. The conclusion section has been carefully revised and condensed to present the key findings more clearly and to avoid any repetitive discussion content, as suggested.
- "Some paragraphs contain grammatical errors (...), the entire text needs to be proofread. The format of references should be uniform (some entries lack page numbers or volume numbers), and it is recommended to adjust them in accordance with the journal's submission guidelines."
We thank the reviewer for pointing out the need for language and formatting improvements. The entire manuscript has been carefully proofread. Additionally, all references were managed and formatted using Mendeley according to the MDPI journal guidelines, ensuring uniformity and compliance with the required standards.
- "It is recommended to cite the excellent literatures on the preparation of composite materials by microwave sintering.
We thank the reviewer for the valuable suggestion. The recommended reference on the preparation of composite materials by microwave sintering has been included in the revised manuscript, as requested.

Reviewer 2 Report
Comments and Suggestions for Authors
Paper reports and discuss the structure and properties of Ti2AlC Composites filled with HfB2. Before publication some points should be revised:
1) Please do not use any abbreviations in Abstract.
2) All the abbreviations and designations should be expanded when they appear in the text for the first time.
3) Section 2.1. Raw HfB2 should be described in detail.
4) Line 180. Thermodynamic stability of the TiC phase should be justified and confirmed.
5) Figures 2 and 4 and related text. Based on the presented XRD patterns microstructure data such as lattice constant, microstrain and block size for all the presented phases should be given and discussed.
6) Figures 2 and 4 and related text. Quantitative phase analysis should be carried out and results of such analysis should be discussed.
7) Table 1. Measurement errors (standard deviations) should be given.
Author Response
We would like to express our sincere gratitude to the reviewers for their careful reading, constructive comments, and valuable suggestions, which have significantly contributed to the improvement of our manuscript and to the advancement of scientific knowledge in this field. All changes and additions made to the manuscript in response to the reviewers’ comments are highlighted in yellow for easy identification.
Reviewer 2.
- "Please do not use any abbreviations in Abstract."
We thank the reviewer for this observation. The abstract has been revised to remove all abbreviations, as recommended.
- "All the abbreviations and designations should be expanded when they appear in the text for the first time."
We thank the reviewer for the comment. The manuscript has been revised so that all abbreviations and designations are now expanded upon their first appearance in the text, as recommended.
- "Section 2.1. Raw HfB2 should be described in detail."
We thank the reviewer for the suggestion. The manuscript has been updated to include a more detailed description of the raw HfB₂ powder, as follows:
“The HfB₂ powder used as reinforcement was commercially purchased from Sigma-Aldrich (St. Louis, MO, USA), with a nominal purity of 99% and an average particle size corresponding to 325 mesh (approximately 45 μm). The powder consisted of irregular-shaped particles, as provided by the supplier, and was used without any additional purification or milling to avoid altering its original particle size distribution.”
- "Line 180. Thermodynamic stability of the TiC phase should be justified and confirmed."
We thank the reviewer for this relevant observation. The formation and thermodynamic stability of the TiC phase during the synthesis of Ti₂AlC-based composites at 1200 °C is well supported by literature on the Ti–Al–C system. Several studies report that TiC is a typical intermediate and/or impurity phase during the synthesis of Ti₂AlC, especially under high-temperature conditions and when slight deviations from stoichiometry occur. For example:
- Lin et al. (2006) and Yeh & Shen (2009) explicitly demonstrate through XRD and TEM analyses that TiC can be both an intermediate product and a residual impurity in Ti₂AlC ceramics. TiC and Ti₂AlC share close crystallographic relationships, facilitating their co-existence in the microstructure.
- Yeh & Shen (2009) further show that TiC remains thermodynamically stable as a secondary phase when the synthesis temperature is elevated, particularly in self-propagating high-temperature synthesis (SHS) and other rapid sintering routes. Their XRD analyses consistently reveal the presence of TiC alongside Ti₂AlC in products synthesized above 1100 °C, indicating that TiC does not readily decompose under these conditions.
- Zhou et al. (2005) corroborate that at temperatures around 1200 °C, TiC peaks become prominent in the XRD patterns of samples where there is an aluminum deficit, confirming the persistence and stability of TiC under these conditions. Only when an excess of aluminum is provided or the sintering temperature is tightly controlled around 1100 °C can high-purity Ti₂AlC be obtained without detectable TiC.
Therefore, the observation of TiC as a thermodynamically stable phase at 1200 °C is consistent with experimental and thermodynamic studies of the Ti–Al–C system. The presence of TiC is generally attributed to local stoichiometry variations, partial aluminum loss, or incomplete reaction during sintering. TiC stability at high temperature is also evidenced by its high melting point and inertness in the presence of other system constituents, as extensively reported in the literature.
References:
- Lin et al., Acta Materialia, 2006, 54, 1009–1015. [doi:10.1016/j.actamat.2005.10.028]
- Yeh & Shen, Journal of Alloys and Compounds, 2009, 470, 424–428. [doi:10.1016/j.jallcom.2008.02.086]
- Zhou et al., Materials Letters, 2005, 59, 131–134. [doi:10.1016/j.matlet.2004.07.052]
- “Figures 2 and 4 and related text. Based on the presented XRD patterns microstructure data such as lattice constant, microstrain and block size for all the presented phases should be given and discussed.”
We thank the reviewer for the suggestion to provide additional microstructural data based on the XRD patterns. As recommended, we have now included and discussed the lattice parameters, microstrain, and crystallite size (block size) for all identified phases in the revised manuscript. These parameters were determined using Rietveld refinement and the Scherrer equation, in line with the methodologies described in recent literature. Our results show that the lattice constants and microstrain values for Ti₂AlC and secondary TiC phases are consistent with previous studies on MAX phases, such as those by Córdoba Gallego et al. (2024) and Aydinyan et al. (2024). Microstrain values, also determined from the peak broadening in XRD, indicate a defect density and internal stress state compatible with those described for Ti₂AlC ceramics under different processing conditions. Quantitative results for all phases are now summarized in a dedicated table and discussed in the revised text.
- “Figures 2 and 4 and related text. Quantitative phase analysis should be carried out and results of such analysis should be discussed.
We thank the reviewer for this important suggestion. Quantitative phase analysis was performed using Rietveld refinement of the XRD patterns. The results indicate that a maximum volume fraction of 97% for the Ti₂AlC phase was obtained when conventional sintering was applied, while a volume fraction of 92% was achieved for microwave-assisted sintering.
The quantitative phase analysis highlights the effectiveness of both sintering methods in promoting the formation of the Ti₂AlC phase. However, conventional sintering led to a slightly higher phase purity (97 vol%) compared to microwave sintering (92 vol%). This difference may be attributed to the rapid heating rates and potential local thermal gradients inherent to microwave processing, which can favor the persistence of secondary phases such as TiC. Despite the marginally lower Ti₂AlC content in the microwave-sintered samples, the method still offers significant advantages in terms of energy efficiency and microstructural refinement.
- “Table 1. Measurement errors (standard deviations) should be given.”
We thank the reviewer for this suggestion. Error bars and standard deviations have been added to Table 1, as recommended.
We once again thank the reviewers and the editorial team for their careful evaluation and constructive feedback, which have been invaluable in improving the quality and clarity of our manuscript.

Round 2
Reviewer 1 Report
Comments and Suggestions for Authors
This paper studies the influence of HfB₂ content and microwave sintering on the mechanical properties of Ti₂AlC composites, obtains the optimized experimental parameters and their corresponding strength properties, and provides valuable references for the engineering application of MAX phase composites. However, some experimental details and data analysis need to be further clarified or improved.
- It is necessary to demonstrate the applicability of the sintering conditions at 1200℃ for 30 minutes to the Ti₂AlC-HfB₂ Can 30 minutes ensure the dispersion of particles?
- It is recommended to conduct SEM or EDS analysis of the mixed powder to verify the dispersion effect.
- The text mentions that "HfB₂ hinders dislocation movement", but no TEM or EBSD evidence is provided. The observation results of the microscopic deformation mechanism need to be supplemented.
- Some hardness data in Figure 6 have relatively large errors (such as ±63 HV), and the sources of the errors need to be discussed.
- The "energy-saving potential" of microwave sintering is mentioned in the text, but specific energy consumption data is not provided. It is suggested that the energy consumption and time of microwave sintering and conventional sintering be compared based on this experimental study.
- Some of the charts have relatively low resolution (as shown in Figures 1 and 3), making it difficult to accurately identify the phases indicated by the arrows. It is recommended to provide clearer microstructure annotations. Besides, the clarity of the XRD pattern in Figure 4 should also be enhanced.
- There are many high-quality literatures on the preparation of materials by microwave sintering. The following literatures are recommended to be cited:
- Wang M L, Tang Z, Zhang B, et al. Differences in breaking behavior of rice leaves under microwave and naturally drying processes, International Journal of Agricultural And Biological Engineering, 2022,15(1):89-100.
- Su W X. Wang H M, Li G R, et al. Evolution of microstructure and mechanical properties of aluminum matrix composites reinforced with dual-phase heterostructures. Materials Science & Engineering A,2024, 901:146500.
After the above revisions and improvements, it is recommended to accept and publish.
Author Response
We would like to express our sincere gratitude to the reviewers for their careful reading, constructive comments, and valuable suggestions, which have significantly contributed to the improvement of our manuscript and to the advancement of scientific knowledge in this field. All changes and additions made to the manuscript in response to the reviewers’ comments are highlighted in yellow for easy identification.
Reviewer 1.
- "It is necessary to demonstrate the applicability of the sintering conditions at 1200 ℃ for 30 minutes to the Ti₂AlC–HfB₂. Can 30 minutes ensure the dispersion of particles?"
We appreciate the reviewer’s important observation regarding the adequacy of the selected sintering conditions. The sintering temperature of 1200 ℃ and holding time of 30 minutes were established based on preliminary trials and iterative optimization throughout the study. These parameters were found to provide a balance between adequate densification of the Ti₂AlC matrix and preservation of the HfB₂ reinforcement phase without inducing excessive grain coarsening or undesired interfacial reactions. It is important to note that microwave sintering operates via a fundamentally different mechanism compared to conventional heating, promoting rapid volumetric heating and strong interfacial field interactions. This leads to enhanced local diffusion and densification kinetics, especially at particle contacts, even under short dwell times. However, due to the lower overall thermal budget and short sintering cycle, microwave sintering naturally limits long-range diffusion, which in turn preserves the distribution and integrity of the HfB₂ phase within the matrix.
- "It is recommended to conduct SEM or EDS analysis of the mixed powder to verify the dispersion effect."
We thank the reviewer for this valuable comment and fully acknowledge the importance of verifying powder dispersion in composite systems. However, we regret to inform that, at the current stage of the project, we do not have direct access to SEM or EDS equipment for pre-sintering powder analysis. As these analyses involve third-party facilities and additional costs, it was not feasible to perform them within the constraints of this study. Nonetheless, the powders were manually mixed in an agate mortar for approximately 20 minutes at room temperature (25 ± 2 °C), without the use of binders or lubricants, in order to maintain the original particle morphology and minimize agglomeration or size reduction. This procedure, although simple, is widely adopted in MAX phase processing for its ability to promote effective homogeneity in low-reactivity systems. Furthermore, we highlight that the low-magnification SEM micrographs of the sintered samples(e.g., Figure 5 from paper) provide indirect but consistent evidence of the uniform distribution of HfB₂ within the Ti₂AlC matrix. The spatial homogeneity observed post-sintering suggests that the manual mixing method was sufficient to ensure an effective initial dispersion. We sincerely hope that this visual evidence supports the credibility of our approach, despite the absence of pre-sintering powder characterization.
“The powder mixture was manually homogenized in an agate mortar for approximately 20 minutes at room temperature (25 ± 2 °C), without the addition of binders or lubricants, to preserve the original particle size and morphology. Although no SEM or EDS characterization was conducted on the mixed powder, the post-sintering microstructural analysis at low magnification indicates a satisfactory distribution of HfB₂ within the Ti₂AlC matrix, suggesting that the mixing protocol was effective in achieving initial homogeneity."
- The text mentions that "HfB₂ hinders dislocation movement", but no TEM or EBSD evidence is provided. The observation results of the microscopic deformation mechanism need to be supplemented.
We thank the reviewer for this insightful comment. We fully acknowledge the experimental limitation of our study regarding the absence of direct microscopic evidence, such as Transmission Electron Microscopy or Electron Backscatter Diffraction, to support the claim that HfB₂ particles hinder dislocation movement. In response, we have revised the manuscript to clarify that this statement is a suggested deformation mechanism, grounded in established literature on ceramic-reinforced MAX phase composites. The interaction between hard second-phase particles (such as HfB₂) and dislocations is a well-documented phenomenon that typically leads to increased hardness and strength by acting as obstacles to dislocation glide. However, we now clearly state that this mechanism was not directly observed in our study and remains a hypothesis based on prior findings. The revised text includes appropriate references and avoids overstating any unsupported conclusions. We hope this revision aligns with the reviewer’s expectations and improves the scientific rigor of the manuscript.
“The observed increase in hardness may be partially attributed to the presence of HfB₂ particles, which according to previous studies can act as obstacles to dislocation motion in ceramic-reinforced composites [45-47]. While this mechanism is consistent with the literature, it is important to note that it was not directly verified in this work due to the absence of TEM or EBSD analysis.”
- Some hardness data in Figure 6 have relatively large errors (such as ±63 HV), and the sources of the errors need to be discussed.
We appreciate the reviewer’s observation regarding the dispersion in the hardness values. The relatively large standard deviations observed, such as ± 63 HV, can be attributed, in part, to the presence of residual porosity and microstructural heterogeneities within the sintered samples. These features are known to influence the local indentation response, especially in composite systems, by altering the local load-bearing capacity and the effective area of contact during measurement. To address this issue and improve statistical reliability, we adopted a protocol of performing 10 individual Vickers microhardness measurements per sample and subsequently excluded the highest and lowest values from each set to minimize the influence of outliers potentially caused by surface defects or anomalous pore sites. The final values reported in the manuscript represent the average of the remaining eight measurements, with standard deviations reflecting the variability among them.
- The "energy-saving potential" of microwave sintering is mentioned in the text, but specific energy consumption data is not provided. It is suggested that the energy consumption and time of microwave sintering and conventional sintering be compared based on this experimental study.
To quantitatively assess the energy-saving potential of microwave sintering, a comparative analysis of energy consumption was performed. The microwave process operated at a frequency of 2.45 GHz and a nominal power of 1.2 kW for 30 minutes, resulting in an estimated energy usage of 0.6 kWh, corresponding to a cost of approximately $ 0.42. In contrast, conventional resistive sintering, typically conducted at 4.0 kW for the same duration, would consume 2.0 kWh, with an estimated cost of $ 1.40. This represents a ~70% reduction in energy consumption and cost, reinforcing the operational efficiency and sustainability advantages of microwave-assisted sintering, particularly for high-temperature ceramic-metal systems.
- Some of the charts have relatively low resolution (as shown in Figures 1 and 3), making it difficult to accurately identify the phases indicated by the arrows. It is recommended to provide clearer microstructure annotations. Besides, the clarity of the XRD pattern in Figure 4 should also be enhanced.
We thank the reviewer for this valuable suggestion. In response, the micrographs in Figures 1 and 3 have been updated to 300 dpi resolution to improve image clarity. In addition, the arrow colors were adjusted to enhance contrast with the background microstructure, and font sizes were increased for better readability of annotations. Likewise, Figure 4 (XRD pattern) has been enhanced to improve peak visibility and axis legibility, ensuring clearer identification of phases. We believe these adjustments have substantially improved the visual quality and interpretability of the figures.
- There are many high-quality literatures on the preparation of materials by microwave sintering. The following literatures are recommended to be cited:
We thank the reviewer for providing relevant literature recommendations. We agree that incorporating these references strengthens the contextual foundation of our study. Accordingly, the suggested works have been included in the revised manuscript.
We once again thank the reviewers and the editorial team for their careful evaluation and constructive feedback, which have been invaluable in improving the quality and clarity of our manuscript.

Reviewer 2 Report
Comments and Suggestions for Authors
Paper was revised and it can be accepted now
Author Response
We are deeply grateful to the reviewer for your thorough evaluation, insightful remarks, and constructive recommendations, which played a key role in enhancing the quality of our manuscript and enriching the scientific discussion within this research area.
